# Impact of Recovery from Febrile Neutropenia on Intra-Individual Variability in Vancomycin Pharmacokinetics in Pediatric Patients

**DOI:** 10.3390/antibiotics14060570

**Published:** 2025-06-02

**Authors:** Yukie Takumi, Ryota Tanaka, Motoshi Iwao, Ryosuke Tatsuta, Hiroki Itoh

**Affiliations:** Department of Clinical Pharmacy, Oita University Hospital, 1-1 Idaigaoka, Hasama-machi, Yufu, Oita 879-5593, Japan; y-sato@oita-u.ac.jp (Y.T.); iwamotoshi@oita-u.ac.jp (M.I.); tatsuta@oita-u.ac.jp (R.T.); itoh@oita-u.ac.jp (H.I.)

**Keywords:** febrile neutropenia, vancomycin, intra-individual, therapeutic drug monitoring

## Abstract

Background/Objectives: The pharmacokinetics of vancomycin (VCM) in patients with febrile neutropenia (FN) are highly variable due to coexisting conditions such as systemic inflammatory response syndrome and augmented renal clearance. Upon hematopoietic recovery, VCM clearance (CLvcm) is expected to normalize, which contributes to intra-individual variability. This study aimed to investigate the factors contributing to intra-individual variability in CLvcm among pediatric patients with FN. Methods: This retrospective, single-center study analyzed 33 pediatric patients (48 FN episodes) who met the inclusion criteria. CLvcm was estimated using Bayesian estimation based on the pediatric population pharmacokinetic model developed by Le et al., and standardized with allometrically scaled body weight. The change (Δ) in each clinical laboratory parameter or CLvcm was calculated as the difference between the values at the current and previous TDM within the same episode. Results: A total of 155 VCM TDM data points were analyzed. Intra-individual comparisons revealed that CLvcm decreased significantly in patients recovering from FN to a non-FN state (*n* = 18, *p* = 0.0285). Further analysis of intra-individual variability revealed that Δ CLvcm correlated significantly with Δ hemoglobin, Δ C-reactive protein, and Δ maximum daily body temperature, with the strongest correlation observed for Δ maximum daily body temperature (rs = 0.325, *p* = 0.001). Multivariate analysis confirmed Δ maximum daily body temperature as a significant factor influencing Δ CLvcm (B = 0.376, 95% CI: 0.074 to 0.678, *p* = 0.015). Conclusions: Maximum daily body temperature was identified as a factor influencing intra-individual variability in CLvcm in pediatric FN patients, particularly during the recovery process from FN to a non-FN state. The finding suggests that dose adjustment based on maximum daily body temperature may allow safe and effective VCM therapy in FN patients.

## 1. Introduction

Febrile neutropenia (FN) is a frequent and clinically relevant complication in patients receiving cytotoxic chemotherapy for cancer, characterized by the presence of both fever and neutropenia. According to the guidelines of the Infectious Diseases Society of America (IDSA), fever is defined as a single oral temperature of ≥38.3 °C or a sustained temperature of ≥38.0 °C for more than one hour. Neutropenia is defined as an absolute neutrophil count (ANC) of <500 cells/mm^3^ or an ANC expected to decrease to <500 cells/mm^3^ within the next 48 h [1]. The Japanese Society of Medical Oncology defined FN as a condition characterized by a neutrophil count less than 500/μL or a predicted decrease to below 500/μL within 48 h, accompanied by fever of axillary temperature ≥ 37.5 °C [2].

FN is primarily induced by cytotoxic anticancer agents (e.g., anthracyclines, cisplatin, ifosfamide) administered to patients with hematologic or solid malignancies [3]. These agents impair both bone marrow hematopoiesis and gastrointestinal mucosal integrity. Consequently, they increase susceptibility to invasive infections caused by translocation of commensal bacteria or fungi across the compromised intestinal barrier. In neutropenic patients, classical signs of inflammation are often absent, and fever is frequently the earliest and sometimes the only indicator of infection. Severe cases of FN can progress to sepsis or septic shock, with in-hospital mortality rates reaching as high as 50% [4]. Therefore, early recognition of FN and prompt initiation of empirical broad-spectrum antimicrobial therapy are crucial to prevent disease progression and reduce the risk of sepsis-related mortality [1].

FN is a condition associated with a high risk of morbidity and mortality; therefore, empirical administration of an anti-pseudomonal agent is recommended before the causative pathogen is identified. If the initial response is inadequate, vancomycin (VCM), an anti-methicillin-resistant *Staphylococcus aureus* (MRSA) agent, is additionally administered [1]. VCM is a highly polar, water-soluble glycopeptide antibiotic that is primarily eliminated via glomerular filtration [5]. The pharmacodynamic efficacy of VCM is best described by the ratio of area under the concentration–time curve (AUC) to minimum inhibitory concentration (MIC), and a target AUC/MIC of 400–600 mg·h/L is recommended for adults to optimize both efficacy and safety [6,7]. In pediatric populations, maintaining AUC below 800 mg·h/L and trough concentration below 15 mg/L has been suggested to be the optimal strategy to minimize AKI risk [6]. Given these considerations, therapeutic drug monitoring (TDM) is recommended to ensure both safety and efficacy of VCM therapy.

In patients with systemic inflammatory response syndrome, the presence of capillary leakage, reduced protein binding, aggressive intravenous fluid resuscitation, and vasopressor administration can increase the volume of distribution (Vd) and enhance renal clearance, leading to decreased plasma concentrations of hydrophilic antibiotics [8]. This phenomenon, known as augmented renal clearance (ARC), is particularly relevant for VCM due to its hydrophilic nature and predominant renal elimination [5]. Indeed, previous studies have reported that achieving target VCM concentrations is often challenging in patients with ARC [9]. ARC has been observed in FN cases, particularly in pediatric FN patients, in whom ARC accelerates the clearance of renally excreted drugs [10,11]. In addition, FN has been associated with lower VCM concentrations in plasma [12].

Upon hematopoietic recovery and transition from the FN to non-FN state, the aforementioned pharmacokinetic alterations are expected to normalize. Consequently, FN patients exhibit marked intra-individual variability in VCM clearance (CLvcm), leading to frequent discrepancies between predicted and observed plasma concentrations in clinical practice. However, no studies to date have systematically investigated the factors influencing intra-individual variability in CLvcm in FN patients.

Given the above background, the objective of this study was to investigate the factors influencing intra-individual variability in CLvcm in pediatric patients with FN.

## 2. Results

### 2.1. Patient Characteristics

The flow of patient selection is shown in Figure 1. As a result of patient selection, a total of 33 eligible FN patients with 48 episodes of FN and 155 VCM TDM data that satisfied the selection criteria were included in the final analysis. All 155 TDM samples were trough concentrations obtained immediately before the next scheduled dose. Table 1 presents the demographic and clinical characteristics of the subjects at the initiation of VCM administration. A total of 48 FN episodes met the inclusion criteria. The male-to-female ratio was 5:6. The median age at VCM initiation was 7.98 years, the median duration of VCM administration was 12 days, and the median time from VCM initiation to the first TDM was 3 days. Because the subjects were FN patients, the maximum daily body temperature and C-reactive protein (CRP) level were high, while hematological parameters were markedly low. The median estimated glomerular filtration rate (eGFR) was markedly high at 175.8 mL/min/1.73 m^2^ due to ARC. Thirty-eight patients (79.2%) had an underlying hematologic malignancy. Blood cultures were positive in 16 patients (33.3%), and *Streptococcus* spp. was the most frequently detected pathogen (six cases, 37.5%).

### 2.2. Correlation Between CLvcm and Clinical Laboratory Parameters at All Data Points

The correlation between CLvcm and various laboratory parameters was analyzed using all available VCM TDM data (*n* = 155). CLvcm was found to correlate positively with total bilirubin (Spearman’s rank correlation coefficient [rs] = 0.287, *p* < 0.001), maximum daily body temperature (rs = 0.260, *p* = 0.001), and eGFR (rs = 0.461, *p* < 0.001); and correlate negatively with neutrophil count (rs = −0.290, *p* < 0.001) and platelet count (rs = −0.226, *p* = 0.005) (Table 2).

### 2.3. Identification of Factors Associated with CLvcm by Multivariate Analysis

To identify factors influencing CLvcm, multiple regression analysis was performed with CRP, neutrophil count, and maximum daily body temperature as independent variables (selection of independent variables is explained in Materials and Methods) (Table 3). Maximum daily body temperature was identified as the significant factor influencing CLvcm (partial regression coefficient [B] = 0.442, 95% confidence interval [CI]: 0.12 to 0.764, *p* = 0.008). In contrast, CRP (*p* = 0.911) and neutrophil count (*p* = 0.746) showed no significant association with CLvcm.

### 2.4. Intra-Individual Comparison of CLvcm Between FN and Non-FN States

Even in patients who met the criteria for FN at the initiation of VCM, some recovered to a non-FN state as a result of the effective antimicrobial therapy and hematologic recovery. Therefore, we evaluated the impact of FN recovery on CLvcm. FN recovery was defined as the transition from an FN state (meeting the FN criteria) to a non-FN state in an individual patient. A total of 18 patients exhibited FN recovery. The mean CLvcm values in 18 paired FN and non-FN states were compared and are presented as mean ± standard deviation. The mean CLvcm in the FN state was significantly higher than that in the non-FN state (7.29 ± 3.01 vs. 5.41 ± 1.89 L/h, *p* = 0.0285, paired *t*-test; Figure 2).

### 2.5. Correlation Between Intra-Individual Changes in CLvcm and in Laboratory Parameters

To explore factors influencing intra-individual variability in CLvcm, the correlation between Δ CLvcm and the Δ value of each laboratory parameter was analyzed using 107 data points. Δ CLvcm showed a negative correlation with Δ hemoglobin (rs = −0.217, *p* = 0.025) and a positive correlation with Δ CRP (rs = 0.194, *p* = 0.046) and Δ maximum daily body temperature (rs = 0.325, *p* = 0.001) (Table 4). Among the laboratory parameters, the correlation coefficient with Δ maximum daily body temperature was the highest.

### 2.6. Identification of Factors Associated with Intra-Individual Δ CLvcm by Multivariate Analysis

To identify factors influencing intra-individual variability in CLvcm, multiple regression analysis was performed using Δ CRP, Δ neutrophil count, and Δ maximum daily body temperature as independent variables. Δ maximum daily body temperature (B = 0.376, 95% CI: 0.074 to 0.678, *p* = 0.015) was identified as the significant factor influencing Δ CLvcm (Table 5). In contrast, Δ CRP (*p* = 0.329) and Δ neutrophil count (*p* = 0.756) showed no significant association with Δ CLvcm.

## 3. Discussion

This study is the first to focus on intra-individual variability in VCM pharmacokinetics in FN patients. The results showed that CLvcm decreased as patients recovered from the FN to the non-FN state, demonstrating large intra-individual variability. Furthermore, multiple regression analysis identified the change in maximum daily body temperature as a significant factor influencing Δ CLvcm in FN patients, suggesting that CLvcm fluctuates in an individual patient depending on changes in body temperature.

Several studies have examined factors associated with the inter-individual variability in CLvcm in pediatric patients [13,14]. These studies have identified body temperature as a significant factor, which is consistent with our findings. A population pharmacokinetic analysis of VCM in 52 pediatric patients who had cardiac arrest reported that those who underwent hypothermia therapy (32–34 °C) exhibited up to a 25% reduction in CLvcm compared to those with normothermia (36.3–37.6 °C) [13]. In addition, a population pharmacokinetic analysis of pediatric patients who received VCM for FN following hematopoietic stem cell transplant identified four covariates for CLvcm: body weight, postmenstrual age, eGFR, and body temperature [14]. The model estimated that CLvcm was 12% higher in patients with a fever of 38 °C or higher than in those below 38 °C. Based on their model, the starting dose recommended for pediatric FN patients with a body temperature below 38 °C was 10% lower than that for FN patients with a temperature of 38 °C or higher. Although these previous studies focused on inter-individual variability, their findings on VCM dosing regimens are also relevant to intra-individual variability.

In the present study, although CRP and neutrophil count were significantly correlated with CLvcm in univariate analyses, they did not remain significant factors in the multivariate model. This suggests that CRP and neutrophil count may not directly influence CLvcm, but rather act as confounding factors that fluctuate concurrently during FN. The associations observed in the univariate analysis are therefore considered to reflect physiological changes associated with the FN state, rather than representing independent contributors to CLvcm variability.

Other factors influencing CLvcm in pediatric patients include renal function, protein binding rate, fluid balance, and concomitant medications [15]. Regarding renal function, FN can induce ARC to increase CLvcm [14,16,17]. In this study, the mean eGFR (mL/min/1.73 m^2^) of FN patients was 175 mL/min/1.73 m^2^. Although there are no clear criteria for ARC in pediatric patients, previous reports have defined ARC in pediatric patients as eGFR above 160 mL/min/1.73 m^2^ [14,16,18]. Our findings showed a positive correlation between CLvcm and eGFR, consistent with these past reports. However, eGFR did not contribute to the intra-individual variability of CLvcm. The present study calculated eGFR using the revised Schwartz equation, which uses creatinine level as the renal function marker. Compared to other renal biomarkers, serum creatinine has been reported to increase with a delay following kidney injury. For example, a study investigating AKI in critically ill adult patients demonstrated that creatinine levels increased 1 to 2 days later than cystatin C levels in patients who developed AKI [19]. In our study, the median time interval for calculating changes in CLvcm or laboratory parameters was 2 (interquartile range: 2–4) days, which is relatively short. Consequently, renal function markers based on creatinine may not have accurately and sensitively reflected renal function. Furthermore, previous studies have pointed out that creatinine-based estimated creatinine clearance or eGFR is not suitable for assessing renal function in ARC cases [11,20,21,22]. These findings suggest that a more sensitive renal function marker may be required to detect short-term intra-individual renal function fluctuations.

According to the free drug hypothesis, only unbound drugs are subject to clearance. Therefore, the VCM protein binding rate may influence CLvcm. However, the present study showed that serum albumin levels did not affect either inter-individual or intra-individual variability in CLvcm. Generally, VCM protein binding is reported to be 50–55% [23]. In critically ill pediatric patients, VCM protein binding has been reported to be lower and more variable than in non-critically ill adults [24,25]. Specifically, younger children (1 month to 5 years) had significantly lower unbound drug concentrations than older children (6–17 years) [26]. While these reports suggest that protein binding of VCM may influence CLvcm in pediatric patients, another study showed no effect of VCM protein binding on the achievement of AUC24 > 400 μg·h/mL [26]. Additionally, although the unbound fraction of VCM correlated negatively with serum albumin levels and correlated positively with renal CL, there was no correlation with total body CL or Vd [27]. These reports suggest that the influence of albumin levels on CLvcm is minimal, which is consistent with the findings of the present study.

In this study, the impact of fluid balance on VCM pharmacokinetics was not evaluated. However, fluid balance may influence the pharmacokinetics of VCM. A study conducted in pediatric patients admitted to the pediatric intensive care unit found that children with negative fluid balance before VCM administration exhibited higher peak and trough concentrations on days 1 and 3 compared to children with positive fluid balance [28]. Additionally, the Vd of VCM was lower on days 1 and 3 in patients with negative fluid balance. These findings suggest that fluid balance may influence changes in the Vd, which warrant further investigations in our study.

Concomitant medications may also influence the pharmacokinetics of VCM. VCM is often co-administered with nephrotoxic drugs, and cases of nephrotoxicity have been reported [29]. In critically ill pediatric patients receiving continuous VCM infusion, those who developed nephrotoxicity exhibited elevated VCM trough levels, and aminoglycosides have been considered a contributing factor [30]. Additionally, cyclosporine co-administration has been reported to affect CLvcm [31]. These studies suggest that the primary mechanism by which concomitant medications affect VCM pharmacokinetics is through their impact on renal excretion, which is the main elimination pathway of VCM. However, in our study, cases of FN with renal impairment were excluded; therefore, the influence of concomitant drugs is considered negligible in this analysis.

Numerous reports have investigated the inter-individual variability in VCM pharmacokinetics, and intra-individual variability is also expected to occur via similar mechanisms. Since VCM has a narrow therapeutic window, TDM is recommended. Particularly in FN patients receiving VCM, ARC can cause dramatic physiological changes, potentially leading to fluctuations in VCM pharmacokinetics [9,14,16,17]. Focusing on intra-individual variability in VCM may achieve appropriate TDM timing and dose adjustment, ultimately improving treatment efficacy and safety. In our study, the change in maximum daily body temperature was identified as a significant factor influencing intra-individual CLvcm. Particularly in patients recovering from the FN to non-FN state, performing TDM when the change in body temperature is marked may help capture the timing of significant CLvcm fluctuation, and appropriate dose adjustment may prevent the abrupt increase in trough level and associated nephrotoxicity.

There are several limitations in this study. First, while the IDSA guidelines define fever based on oral temperature measurements, axillary temperature was used in this study. Although rectal temperature is generally considered more accurate than oral temperature, it is invasive and uncomfortable, making it difficult to obtain in children other than infants. Furthermore, pediatric patients with FN often present with thrombocytopenia, which increases the risk of bleeding and makes rectal measurements less appropriate. In Japanese pediatric clinical settings, axillary temperature is more commonly used than oral temperature due to its safety and practicality. As this study was retrospective in nature, rectal temperature data were not available. This discrepancy in the method of temperature measurement may have introduced variability in the classification of febrile episodes. Future studies should adopt standardized and consistent temperature measurement methods. Second, the potential influence of antipyretic and steroid use on the results cannot be ruled out, as this study did not assess the use of these medications. If antipyretics or steroids were administered, they may have affected body temperature measurements or altered inflammatory responses, potentially influencing the observed relationship between body temperature and CLvcm. Recording and analyzing medication use should be considered in future studies. Third, clinical laboratory parameters measured on the day of VCM TDM were used to identify factors influencing CLvcm. However, not all the relevant parameters were measured simultaneously with TDM. If important factors influencing CLvcm were among the unmeasured parameters, their impact may not have been assessed accurately, potentially leading to an incomplete understanding of the determinants of intra-individual variability. Future studies should ensure comprehensive and synchronized measurement of laboratory values at each TDM time point. Fourth, when calculating Δ values for laboratory parameters, the intervals between sample collection varied widely, with a median of 2 days (interquartile range: 2–4 days). This variability may have obscured the influence of factors other than body temperature, impeding accurate assessment of their effects on CLvcm. Prospective studies with fixed sampling intervals are needed to reduce variability. Lastly, all concentration measurements used for Bayesian estimation were based on trough sampling. Since peak concentrations were not utilized, the effect of FN on Vd could not be accurately evaluated. The absence of peak data limits the ability to assess potential pharmacokinetic changes beyond clearance variability. Future studies should include peak sampling to fully characterize VCM pharmacokinetics.

## 4. Materials and Methods

### 4.1. Subjects

This retrospective, single-center study was conducted using patient data extracted from the electronic medical record system. A total of 197 pediatric patients with FN (262 episodes) who received VCM in the Department of Pediatrics at Oita University Hospital between October 2013 and November 2022 were initially identified. Of these, 140 patients (153 episodes) who did not meet the criteria for FN at the initiation of VCM therapy were excluded from analysis. FN was defined as an absolute neutrophil count of less than 500/μL, or less than 1000/μL with anticipated decline to below 500 cells/μL within 48 h, in conjunction with fever defined as an axillary temperature of 37.5 °C or above.

Of the 109 episodes (57 patients) that met the inclusion criteria, the following were excluded from the analysis: one episode in a patient aged 19 years or older, 20 episodes without VCM TDM, 37 episodes with only a single TDM, two episodes in patients with chronic kidney disease (CKD), and one episode in a patient who developed AKI during the course of VCM therapy. CKD was defined according to the criteria established by the Japanese Pediatric CKD Study Group as eGFR lower than 60 mL/min/1.73 m^2^ [32]. AKI was defined based on the Kidney Disease: Improving Global Outcomes criteria, with stage 1 or higher classified as AKI [33]. Of the episodes that met the aforementioned selection criteria, several TDM data were excluded due to the following reasons: the time of injection or sampling was unknown, or blood was not collected at the trough sampling time.

### 4.2. Data Collection

Patient background and laboratory data at the time of VCM initiation and TDM data were collected from the electronic medical records. Laboratory data were also obtained on the same day as each TDM sampling. The following parameters were recorded: sex, age, weight, height, body temperature, white blood cell count, neutrophil count, hemoglobin (HGB), platelet count, CRP, albumin, total bilirubin, aspartate aminotransferase, alanine aminotransferase, gamma-glutamyl transpeptidase, serum creatinine (Scr), blood urea nitrogen, and underlying diseases. The eGFR was calculated using the Schwartz formula: eGFR = 0.413 × height (cm)/Scr (mg/dL) [32]. For microbiological tests, a positive result was defined as the detection of any pathogen in a single set of blood cultures. Information regarding VCM administration, including single dose, daily dose frequency, administration time, and total duration, as well as TDM-related data such as sampling time and date, were also collected. The change in each clinical laboratory value and the change in CLvcm (denoted as Δ) were calculated as the difference between the values at the current TDM and the previous TDM [34]. For patients with *n* trough samples, the number of analysis points per patient was *n* − 1, with a total of 107 data points for analysis. The interval between each sampling point had a median of 2 days (interquartile range: 2–4 days).

### 4.3. Pharmacokinetic Analysis

Pharmacokinetic parameters were estimated using a population pharmacokinetic model developed for pediatric patients by Le et al. because it was constructed with extensive and diverse concentration data, as well as a wide range of pediatric ages [35]. In this model, body weight, Scr, and age were incorporated as covariates for CLvcm, while body weight was included as a covariate for Vd. The equations used were as follows:
CLvcm (L/h) = 0.248 × (body weight)^0.75^ × (0.48/Scr)^0.361^ × [ln(age)/7.8]^0.995^
Vd (L) = 0.636 × (body weight)

Based on this population pharmacokinetic model, individual CL values at the sampling points were estimated using Bayesian estimation with the least-squares method [MULTI (BAYES)] from the measured trough concentrations [36]. To account for variability in body size across different pediatric age groups, allometric scaling was applied to normalize CLvcm using body weight [37]:Standardized CLvcm = individual CLvcm/(body weight/70 kg)^0.75^.

### 4.4. Statistical Analysis

Statistical analyses were performed using Predictive Analysis Software (PASW) Statistics version 27 (SPSS Inc., Chicago, IL, USA). Baseline clinical data were expressed as median [interquartile range] for continuous variables and as number (%) for categorical variables. The normality of each variable was assessed using the Shapiro–Wilk test, and the correlation between parameters was analyzed using Spearman’s rank correlation coefficient. For patients who recovered from the FN to the non-FN state and underwent TDM in both conditions, the mean CLvcm values in the two states were calculated. The difference in mean CLvcm between the two states was evaluated using a paired *t*-test.

Factors associated with CLvcm or Δ CLvcm were identified using multiple regression analyses. Based on the underlying mechanisms, body temperature, neutrophil count, and CRP were selected as dependent variables, as they were likely to influence CLvcm [14,38]. Although a correlation between CLvcm and HGB has been reported in septic patients [39], HGB was excluded due to the potential influence of transfusions administered during the FN period. Furthermore, since Scr was incorporated as a covariate in the population model for CL, renal function test values were not included as independent variables. To test the normality of residuals, the Durbin–Watson test and Shapiro–Wilk test were performed. Collinearity and multicollinearity among independent variables were assessed using a variance inflation factor threshold of 10. Selected independent variables were forcibly incorporated into the model based on Schwarz’s Bayesian Information Criterion. A *p*-value below 0.05 was considered statistically significant.

## 5. Conclusions

In pediatric patients with FN, significant intra- and inter-individual variability in VCM pharmacokinetics complicates optimal dosing, potentially leading to reduced therapeutic efficacy and an increased risk of adverse effects. This study identified body temperature during FN as a key factor influencing intra-individual variability in CLvcm. These findings suggest that dose adjustments based on changes in body temperature may enable safer and more effective VCM therapy in pediatric patients with FN.

## Figures and Tables

**Figure 1 antibiotics-14-00570-f001:**
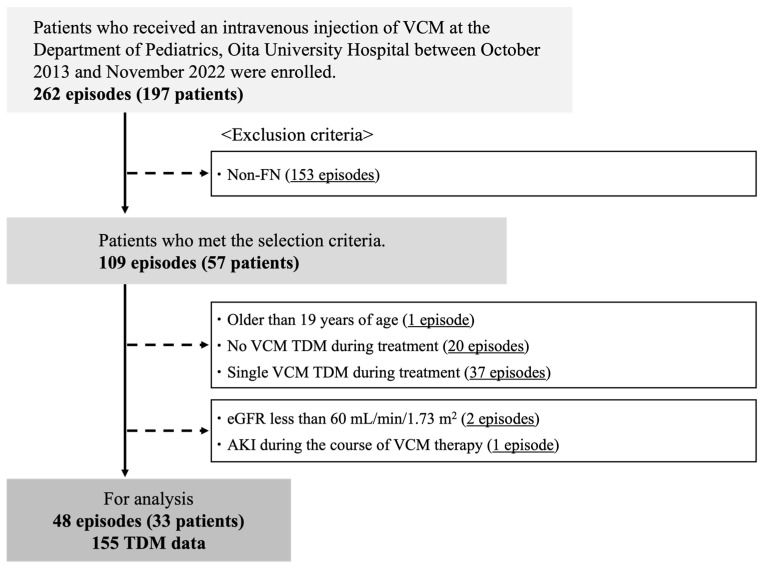
Flowchart of patient selection in the present study. VCM, vancomycin; FN, febrile neutropenia; TDM, therapeutic drug monitoring; eGFR, estimated glomerular filtration rate.

**Figure 2 antibiotics-14-00570-f002:**
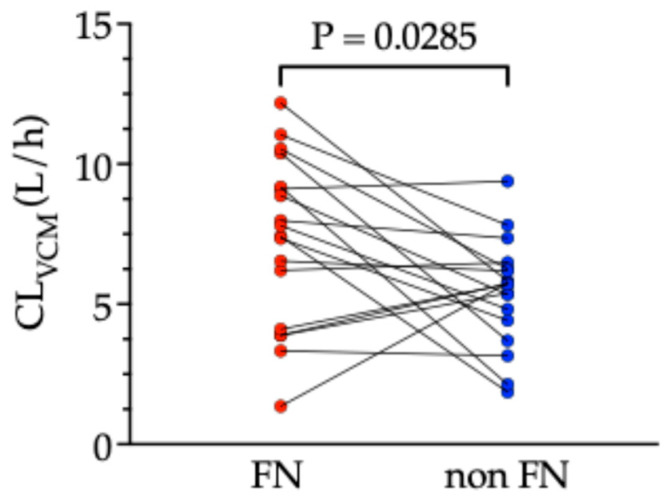
Intra-individual comparison of vancomycin clearance between FN and non-FN states. The difference between the two states was evaluated using a paired *t*-test. CLvcm, vancomycin clearance; FN, febrile neutropenia.

**Table 1 antibiotics-14-00570-t001:** Patient characteristics at the initiation of vancomycin administration.

Characteristics	Value	
Number of episodes; *n*	48	
Sex ratio * (male/female); *n* (%)	15/18	(45/55)
Age (years)	7.98	[3.71–12.2]
Body weight (kg)	25.5	[12.6–37.7]
Height (cm)	119.1	[94–146]
Duration (days)	12	[10.0–14.3]
Vancomycin daily dose (mg/kg)	45.7	[40.7–57.1]
Time to blood sampling for initial therapeutic drug monitoring (days)	3	[2.0–4.0]
Maximum body temperature (°C)	38.4	[38.0–39.0]
White blood cell count (×10^3^/μL)	0.22	[0.11–0.40]
Neutrophil count (/μL)	0	[0–12.6]
Hemoglobin (g/dL)	8.9	[7.48–10.1]
Platelet count (×10^3^/μL)	29	[19.3–44.3]
C-reactive protein (mg/dL)	3.71	[1.03–7.35]
Serum albumin (g/dL)	3.46	[2.99–3.69]
Total bilirubin (mg/dL)	0.65	[0.42–1.02]
Aspartate aminotransferase (U/L)	27.9	[18.9–54.0]
Alanine aminotransferase (U/L)	28.9	[16.5–66.2]
Gamma-glutamyl transpeptidase (U/L)	41.5	[26.6–106.4]
Serum creatinine (mg/dL)	0.28	[0.18–0.35]
Estimated glomerular filtration rate (mL/min/1.73 m^2^)	175.8	[157.4–212.6]
Blood urea nitrogen (mg/dL)	9.4	[6.85–12.4]
No bacteria detection; *n* (%)	32	(66.7)
Bacteria detection in blood culture; *n* (%)	16	(33.3)
*Coagulase-negative Staphylococcus* spp.	4	(25)
*Enterococcus* spp.	1	(6.25)
*Methicillin-resistant Staphylococcus aureus*	1	(6.25)
*Methicillin-sensitive Staphylococcus aureus*	1	(6.25)
*Streptococcus* spp.	6	(37.5)
Others	3	(18.75)
Primary disease; *n* (%)		
Hematological malignancy	38	(79.2)
Solid tumor	8	(16.7)
Others	2	(4.2)

Data are expressed as numbers (%) for categorical variables and median [interquartile range] for continuous variables. * Sex ratio was calculated for 33 patients studied.

**Table 2 antibiotics-14-00570-t002:** Correlation of vancomycin clearance with clinical laboratory parameters.

Parameter	Correlation Coefficient	*p*-Value (Two-Tailed)
C-reactive protein (mg/dL)	0.142	0.077
Serum albumin (g/dL)	0.062	0.458
Total bilirubin (mg/dL)	0.287	<0.001
Aspartate aminotransferase (U/L)	0.026	0.747
Alanine aminotransferase (U/L)	−0.133	0.099
Gamma-glutamyl transpeptidase (U/L)	0.012	0.892
Blood urea nitrogen (mg/dL)	−0.115	0.154
Serum creatinine (mg/dL)	−0.118	0.143
Neutrophil count (/μL)	−0.290	<0.001
Platelet count (×10^3^/μL)	−0.226	0.005
Hemoglobin (g/dL)	−0.071	0.378
Maximum daily body temperature (°C)	0.260	0.001
Estimated glomerular filtration rate (mL/min/1.73 m^2^)	0.461	<0.001

Correlation was analyzed by Spearman’s rank correlation coefficient.

**Table 3 antibiotics-14-00570-t003:** Multivariate analysis for factors associated with vancomycin clearance.

Independent Variable	Parameter				
	*p*-Value	B	95% CI	β	Adjusted R Squared
C-Reactive protein (mg/dL)	0.911	3.01 × 10^−3^	−0.050 to 0.056	0.010	0.047
Neutrophil count (/μL)	0.746	−3.83 × 10^−5^	−2.72 × 10^−4^ to 1.95 × 10^−4^	−0.026	
Maximum daily body temperature (°C)	0.008	0.442	0.12 to 0.764	0.244	

95% CI, 95% confidence interval; B, partial regression coefficient; β, standardized regression coefficient.

**Table 4 antibiotics-14-00570-t004:** Correlation between changes in vancomycin clearance and changes in clinical laboratory parameters.

Parameter	Correlation Coefficient	*p*-Value (Two-Tailed)
Δ C-reactive protein (mg/dL)	0.194	0.046
Δ Serum albumin (g/dL)	−0.084	0.411
Δ Total bilirubin (mg/dL)	0.074	0.450
Δ Aspartate aminotransferase (U/L)	0.003	0.976
Δ Alanine aminotransferase (U/L)	0.025	0.799
Δ Gamma-glutamyl transpeptidase (U/L)	−0.075	0.471
Δ Blood urea nitrogen (mg/dL)	−0.026	0.787
Δ Serum creatinine (mg/dL)	−0.180	0.064
Δ Neutrophil count (/μL)	−0.151	0.122
Δ Platelet count (×10^3^/μL)	0.077	0.433
Δ Hemoglobin (g/dL)	−0.217	0.025
Δ Maximum body temperature (°C)	0.325	0.001
Δ Estimated glomerular filtration rate (mL/min/1.73 m^2^)	0.112	0.250

Correlation was analyzed by Spearman’s rank correlation coefficient.

**Table 5 antibiotics-14-00570-t005:** Multivariate analysis for factors associated with changes in vancomycin clearance.

Independent Variable	Parameter				
	*p*-Value	B	95% CI	β	Adjusted R Squared
Δ C-reactive protein (mg/dL)	0.329	0.029	−0.030 to 0.088	0.097	0.055
Δ Neutrophil count (/μL)	0.756	3.32 × 10^−5^	−1.78 × 10^−4^ to 2.45 ×10^−4^	0.030	
Δ Maximum daily body temperature (°C)	0.015	0.376	0.074 to 0.678	0.245	

95% CI, 95% confidence interval; B, partial regression coefficient; β, standardized regression coefficient.

## Data Availability

The datasets generated and analyzed for this study are available from the corresponding author upon reasonable request.

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
