# Peer review of "Impact of Recovery from Febrile Neutropenia on Intra-Individual Variability in Vancomycin Pharmacokinetics in Pediatric Patients"

_antibiotics, 2025, doi:10.3390/antibiotics14060570_

Round 1

Reviewer 1 Report

Comments and Suggestions for Authors

Authors provided a thorough exploration of febril neutropenia cases at Oita University Hospital in Japan.

The authors paid great attention to the case development and discussion based on the

results. I still have some comments regarding the manuscript.

Overall I believe this manuscript deserves to be published in this prestigious journal.

Comment 1: I believe the numerical data from most of the Tables should be aligned with the heading in their column. Plus, although the information in the Tables is detailed, their presentation could be improved.

Comment 2: when talking about “male-to-female ratio” of patients (lines 92-93), I believe that would be the total number of patients study, the ratio would actually be 5:6. 

Comment 3: Line spacing in the References section is unnecesary.

Comment 4: In my opinion, the Conclusion contains general phrases and needs to be
improved and more specific in order to improved the quality of presentation.

Reviewer 2 Report

Comments and Suggestions for Authors

Choosing the best dose for administration of Vancomycin is challenging due to its marked intra-individual and inter-individual variability. administration in pediatric population, especially with neutropenia is more challenging. This paper add value for practice through the changing in body temperature as a factor influencing intra-individual variability in CLvcm in pediatric patients with febrile neutropenia.

Reviewer 3 Report

Comments and Suggestions for Authors

I have read this paper with great interest, and highly value the effort reported by Takumi et al. for their creativity to use clinical collected samples to explore 'hyperfiltration' in children with febrile neutropenia. 

I still have suggestions to further improve the clarity of the paper

for the abstract, it should be clearer what model has been applied (for the Bayesian estimation). for both the abstract, and the full paper, it should be clearer that the changes were 'in the same episode' ? I assume that this is the case, but should be made more explicit, and perhaps you could add the time duration between both sampling episodes ? 

perhaps there is also value to add the sampling strategy, assuming that through levels were collected ? 

Table 1, administration period, perhaps duration is more accurate

To provide clarity, are all lab vlaues as collected at initiation, or at TDM monitoring, From a mechanistic point of view, this is not very relevant, but it is relevant for potential clinical use. 

Can you also add some 'quantification' to the abstract and full paper ? how much higher/lower (cfr figure 2).

I agree on the reflection on the time dependent limitations on creatinine 'reactivity'. 

A specific argument for non-rectal temperature measurement is that these children generally have thrombocytopenia, so that this 'absence' of rectal measurements is rather 'standard'. 

Why and how have you 'selected' Le et al for the Bayesian exercises. 

Perhaps we need some more context on the Ethics handling. 

Reviewer 4 Report

Comments and Suggestions for Authors

1-Provide references from lines 35 to 43.

2-Authors said in the first line Febrile neutropenia (FN) is defined by the Japanese Society of Medical Oncology (even did not provide reference) and how about other societies globally like ( Infectious Diseases Society of America (IDSA) & National Comprehensive Cancer Network (NCCN) (U.S.), European Society for Medical Oncology (ESMO) & European Conference on Infections in Leukemia (ECIL) (Europe) and American Society of Clinical Oncology (ASCO) & Multinational Association for Supportive Care in Cancer (MASCC) etc,)and what differences NF cutoff values, temperature, and risk stratification. I suggest including a few lines or making a table for broader understanding and differences.

3-In line 39, the authors mentioned that cytotoxic anticancer agents were administered; please provide a few examples of these agents.

4- Table 3: What is the adjusted R-squared value of temp and CRP?

5-  revise and insert references from 231 to 243 lines

6- In the last paragraph, the authors mentioned the study's limitations but did not suggest any possible solution to overcome these limitations.

7—Lines 108-112 showed that CLvcm and various laboratory parameters correlate positively and negatively; how and why should be discussed in the discussion?

8- The conclusion should be improved for readers to understand it better.

9- Please make short sentences; several sentences are too large throughout the manuscript.
